# Dual-Band Multi-Layer Antenna Array with Circular Polarization and Gain Enhancement for WLAN and X-Band Applications

**DOI:** 10.3390/mi16020203

**Published:** 2025-02-10

**Authors:** Bal S. Virdee, Tohid Aribi, Tohid Sedghi

**Affiliations:** 1Center for Communications Technology London, Metropolitan University, London N7 8DB, UK; 2Department of Electrical Engineering, Miandoab Branch, Islamic Azad University, Miandoab, Iran; tohidaribi@gmail.com (T.A.); sedghi.tohid@gmail.com (T.S.); 3Department of Electrical Engineering, Urmia Branch, Islamic Azad University, Urmia, Iran

**Keywords:** antenna, antenna array, circular polarization, microstrip technology

## Abstract

This paper presents a novel multi-layer, dual-band antenna array designed for WLAN and X-band applications, incorporating several innovative features. The design employs a pentagon-shaped radiating element with parasitic strips to enable dual-band operation. A dual-transformed feed network with chamfered feed strip corners minimizes radiation distortion and cross-polarization while introducing orthogonal phase shifts to achieve circular polarization (CP) at the X-band. A Fabry–Pérot structure, strategically placed above the array, enhances gain in the WLAN band. The antenna demonstrates an impedance bandwidth of 1.8 GHz (S_11_ < −10 dB) at the WLAN band, with 36% fractional bandwidth, and 4.3 GHz at the X-band, with 43% fractional bandwidth. Measured peak gains are 7 dBi for the WLAN band and 6.8 dBi for the X-band, with favourable S_11_ levels, omni-directional radiation patterns, and consistent gain across both bands. Circular polarization is achieved within 8.5–10.4 GHz. Experimental results confirm the array’s significant advancements in multi-band performance, making it highly suitable for diverse wireless communication applications.

## 1. Introduction

The rapid proliferation of WLAN devices and the increasing demands for efficient wireless communication systems highlight the necessity for multi-band antenna arrays capable of operating seamlessly across diverse frequency ranges. These systems must be compact, cost-effective, and capable of producing omnidirectional radiation patterns while maintaining robust performance across all operating bands [1,2,3]. Designing such systems presents numerous challenges, particularly in achieving compactness, broadband functionality, and stable radiation patterns without compromising efficiency.

Key factors in multi-band antenna array design include the shape and configuration of resonator elements, ground planes, and slots within the ground plane. These elements play a pivotal role in enabling multi-band functionality and achieving circular polarization (CP) within the desired frequency range [4]. CP is particularly advantageous in modern wireless communication systems because it ensures reliable device performance regardless of the relative orientation of the transmitter and receiver. This characteristic mitigates multipath fading, a common challenge in urban and indoor environments, and enhances overall system reliability. Furthermore, CP elements outperform linearly polarized (LP) elements in adverse weather conditions due to their superior penetration capabilities [5].

To meet the performance demands of multi-band devices, antenna arrays must exhibit broadband impedance matching, high gain, and consistent radiation patterns across their operating frequency bands. Recent advancements in multi-band monopole antenna designs showcase a variety of configurations, ranging from compact to large elements, each targeting broad impedance bandwidths, dual- or multi-frequency modes, and desirable radiation characteristics. However, achieving these objectives often requires integrating additional design features, such as slots, slits, and parasitic elements, into the radiating elements, ground planes, or feed networks [6,7,8,9]. While effective, these approaches can result in increased design complexity and challenges related to manufacturing and implementation.

Another critical consideration in antenna design is gain enhancement. Achieving higher gain often necessitates larger antenna elements or more complex configurations, which may be impractical for compact systems. The Fabry–Pérot (FP) resonator method offers a promising solution by enhancing antenna gain within narrow bandwidths. This approach involves the use of an FP structure to improve the directional focus of the radiation pattern, thus increasing gain. However, traditional FP-based designs and other gain-enhancing topologies often require intricate feeding networks, adding to the overall complexity and cost of the system [10,11,12].

In this context, we propose a novel antenna array designed to address these challenges while maintaining simplicity, compactness, and performance. The proposed design targets dual-band operation, specifically covering WLAN and X-band frequencies. The array features a 4 × 1 configuration of pentagon-shaped patches, which are paired with parasitic resonators to enable multi-band functionality. The array is excited using a simplified half-power feed network that introduces differential phase shifts, enabling circular polarization in the X-band. This mechanism supports dual-band operation while minimizing the complexity of the feed system. Furthermore, the resonance frequencies can be precisely tuned by adjusting the gap between the pentagonal radiators and the parasitic elements, offering flexibility for various application requirements.

To optimize the array’s performance, a step change is introduced to the ground plane of the coplanar waveguide (CPW), which significantly enhances impedance matching. Additionally, a Fabry–Pérot layer is strategically positioned 0.5λo (@5.5 GHz) above the antenna array to boost gain in the WLAN band, ensuring strong and consistent radiation performance across the dual-band spectrum.

This paper demonstrates the versatility and effectiveness of the proposed antenna array, emphasizing its dual-band capabilities, circular polarization performance in the X-band, and the adjustability of its resonance frequencies for specific design needs. The innovative combination of a cooperative feed mechanism, tuneable resonance features, and gain-enhancing structures positions this antenna array as a robust and practical solution for modern wireless communication systems, including applications in WLAN and X-band environments.

### Antenna Array Parasitic Resonator

The proposed antenna array comprises a 4 × 1 configuration of pentagon-shaped patches, each flanked on either side by conformal parasitic resonators, as depicted in Figure 1. These parasitic elements create additional resonances that interact with the primary patch resonance, effectively broadening the overall bandwidth through multiple coupled resonant modes. The array is excited through a half-power feed network, and the ground plane is enhanced with hexagonal slots positioned directly beneath each resonator. The purpose of the hexagonal slots is to disrupt surface waves by introducing periodic discontinuities that limit their propagation. Without these slots, surface waves can cause interference, resulting in higher reflections and reduced radiation efficiency. In addition, closely spaced antenna elements in an array are prone to mutual coupling, which can degrade overall performance. The hexagonal defective ground structure (DGS) mitigates this coupling by modifying the ground current distribution. Consequently, the slots enhance radiation pattern stability, minimize unwanted reflections, and reduce cross-polarization, thereby improving directional radiation performance.

Figure 1a illustrates the Fabry–Pérot layer, consisting of a 2 × 4 array of resonant structures positioned above the antenna array to enhance gain. The steps for realizing the radiating element are detailed in Figure 2 and Figure 3. The radiating structure features a pentagon-shaped patch surrounded by parasitic elements that conform to its geometry. The adjacent ground plane is stepped to form a coplanar waveguide (CPW) structure. This design is built on an FR4 substrate with a thickness of 1 mm, a relative dielectric constant of 4.4, and a loss tangent of 0.024. The antenna is fed through a 50 Ω CPW. The dimensions of the radiating element were optimized using CST Microwave Studio, with the substrate measuring approximately 17 × 17 mm^2^. The pentagon’s top sides are 6.82 mm, and the left and right sides are 9.40 mm each. A gap of 0.7 mm between the patch and the parasitic resonators is critical for optimal electromagnetic coupling and impedance matching. Additionally, proposed modifications to the ground plane further enhance the electromagnetic environment and improve the antenna’s impedance bandwidth.

To achieve optimal impedance matching, additional modifications were made to the ground structure near the feedline. The effects of these changes are illustrated in Figure 2a, which shows the reflection coefficient (S_11_) response as the radiator evolves from an equilateral triangle to a diamond shape and, finally, to a pentagon.

For Antenna #3, the parameters in the table provided in Figure 2b influence the reflection coefficient performance in Figure 2a as follows:The size of the substrate (W_sub_, L_sub_) impacts the overall radiation efficiency and impedance matching. Larger or optimized dimensions help in achieving better electromagnetic coupling between the radiating elements and parasitic structures.The gap (gap, G, Gc) between the radiating patch and parasitic resonators significantly affects the coupling strength and, consequently, the bandwidth and impedance matching. A gap of 0.7 mm was found optimal for enhancing multi-band operation and achieving good reflection performance at target frequencies.The parameters G and Gc relate to spacings in the ground plane and resonator arrangement, which are critical for maintaining the proper impedance bandwidth.Changes to the ground structure (L_gnd_), such as hexagonal slots and step changes in the CPW, improve impedance matching. The inclusion of stepped modifications aligns better with the patch’s resonances, reducing reflections and enhancing S11.Parasitic resonators on both sides of the pentagon are designed to interact electromagnetically with the main radiating element, supporting dual-band functionality by introducing additional resonant frequencies. This is evident in the dual-band reflection curve where the second band (9.8–11.8 GHz) is influenced by these resonators.

These parameters collectively enhance the reflection coefficient performance of Antenna #3, enabling it to achieve two well-defined impedance bandwidths where S11 ≤ −10 dB. This indicates that the antenna is well-matched for efficient power transfer with minimal mismatch losses across two frequency ranges: 5.9–9.0 GHz and 9.8–11.8 GHz, as shown in Figure 2a. In comparison, Antenna #1 demonstrates a single-band impedance bandwidth from 5.75 GHz to 8.9 GHz, while Antenna #2 slightly improves on this, covering a range from 5.9 GHz to 9.2 GHz.

These results underscore the antenna’s versatility across multiple frequency bands. Such adjustments in the ground structure and antenna geometry are essential for achieving the desired impedance bandwidth and multi-band functionality.

The optimal dimensions of the parasitic elements were determined through a parametric study, with key parameters for Antenna #3 displayed in Figure 3b. These optimizations, performed using the CST Microwave Studio, further refined the antenna’s performance, ensuring it met the desired specifications.

## 2. Dual-Band Antenna Array Feed Network

The 4 × 1 antenna array is excited through a power divider and a dual λ/4 transformer setup, as illustrated in Figure 4. The excitation process begins with the initial power divider, which evenly distributes the input signal among the radiating elements. A quarter-wavelength (λ/4) transformer is then employed to introduce a 90-degree phase shift to the split signals. Subsequently, a second λ/4 transformer further divides the excitation power and applies an additional 90-degree phase shift. This sequential phase-shifting process generates circular polarization by creating two orthogonal modes—an essential feature for various applications in antenna technology and wireless communication systems.

Figure 4a shows the simulated response of the feeding network, and Figure 4b shows the 4 × 1 antenna array excited through the feed network. It is evident that the feed network has a minimal effect on the overall performance of the antenna. The simulation results confirm that Antenna #3 effectively operates in dual-band mode, covering both WLAN and X-bands, demonstrating its suitability for modern wireless communication applications.

## 3. Network Fabry–Perot Layer

The Fabry–Pérot (FP) unit cell is specifically designed to resonate at 5.5 GHz to improve the antenna array’s performance in the WLAN frequency band, with a focus on enhancing gain and maintaining a consistent radiation pattern. The unit cell features a microstrip structure composed of superimposed “+”, “×”, and square-shaped transmission lines, as illustrated in Figure 5. Its dimensions are optimized to achieve resonance at 5.5 GHz, which strengthens the directive radiation pattern at this frequency and boosts the antenna’s gain. This design ensures better signal quality and improved coverage within the targeted WLAN band.

The FP layer, consisting of a 2 × 4 unit-cell arrangement, is strategically positioned above the proposed 4 × 1 antenna array (Antenna #3) with an air gap of 0.5λo (@ 5.5 GHz). Maintaining this air gap is crucial, as deviations can result in destructive interference, causing higher reflections, degraded impedance matching, increased S11 values, and reduced power transfer efficiency, which negatively impacts antenna performance. Changes in the gap also affect the antenna’s resonance and bandwidth. An improper gap may shift the FP layer’s resonance frequency, leading to misalignment with the antenna’s operational frequencies and a narrowing of the impedance bandwidth.

The 2 × 4 unit-cell configuration is chosen to balance performance with compactness and practicality. While a larger FP structure might provide a slight improvement, it would also introduce mechanical and spatial challenges, making it unsuitable for compact communication devices. This proof-of-concept design aims to demonstrate the effectiveness of the FP layer while maintaining a practical system size.

The FP layer is positioned slightly off-centre rather than directly above the antenna array’s centre to optimize field distribution and constructive interference at operating frequencies. This placement helps prevent destructive interference, improving both gain and impedance matching without significantly affecting overall performance. The transmission phase is influenced by the FP structure’s alignment with the antenna. Poor positioning or misalignment can lead to phase distortion and performance degradation due to destructive interference. However, when the position and gap are optimized, the FP structure enhances phase coherence, promoting constructive interference and stable performance across the target frequencies.

The primary purpose of the FP structure is to enhance the gain of the antenna array, particularly in the WLAN band, by reinforcing the directive radiation pattern and improving overall efficiency.

## 4. Measured Results

Figure 6 presents photographs of the fabricated antenna array, which has dimensions of 75 × 36 × 1 mm^3^. To enhance the directivity and gain of the proposed antenna array at the WLAN band, a Fabry–Pérot layer, comprising a 4 × 4 unit-cell arrangement, was positioned above the array at a gap equivalent to half a wavelength (27 mm), as shown in Figure 6b.

The reflection coefficient and the corresponding impedance bandwidth of the dual-band circularly polarized antenna array were measured using a Keysight PNA-X Microwave Network Analyzer (N5242A) (Keysight, Wokingham, UK). Figure 6c compares the simulated and measured reflection coefficient responses of the antenna array. A ripple observed in the response is attributed to interactions between the electromagnetic waves radiated by the array and the FP layer. Despite this, there is strong agreement between the simulated and measured results, validating the accuracy of the simulation model.

The deviation observed around 6 GHz is primarily due to manufacturing tolerances and imperfections in the soldering of the SMA connectors. These imperfections cause impedance mismatches and losses that were not accounted for in the simulation.

Figure 6d shows the peak gain of the antenna array with and without the Fabry–Pérot layer. The addition of the FP layer significantly enhances the gain in the WLAN band, particularly within the 5–6 GHz range, achieving a 2 dB improvement. This gain enhancement is attributed to the resonant characteristics of the FP layer in this frequency range. For the X-band, the gain steadily increases from 4.8 dBi at 8 GHz to 6.8 dBi at 12 GHz. This increase results from the shorter wavelengths at higher frequencies, which effectively enlarge the antenna’s aperture relative to the wavelength, enhancing its ability to capture and radiate electromagnetic waves.

To enhance gain in the X-band, the FP unit cells could be modified to support dual-resonant behavior, with a secondary resonance tuned to a frequency within the X-band. This would enable the FP structure to improve gain at both WLAN and X-band frequencies. Additionally, altering the geometry of the FP unit cells could increase their effectiveness at shorter wavelengths.

The measured axial ratio (AR) of the antenna array as a function of frequency is presented in Figure 6e. Circular polarization is characterized by an AR of 3 dB or less. The antenna demonstrates CP in the second operational band, spanning 5.4–6.1 GHz and 8.9–10.4 GHz. Figure 6f,g depicts the measured radiation patterns for left-handed circularly polarized (LHCP) and right-handed circularly polarized (RHCP) waves at φ = 0° and φ = 90°, measured at 5.7 GHz and 9.5 GHz, respectively.

Although the gain improvement achieved with the FP layer is relatively modest, this approach offers a compact and cost-effective method for gain enhancement. Such improvements are particularly valuable in applications where even small increases in gain can have a significant impact on system performance.

The presented work demonstrates advancements compared to previous antenna designs, as summarized in Table 1. Unlike the prior designs [13,14,15], which do not incorporate circular polarization, the proposed antenna achieves CP, a critical feature for applications requiring reliable performance across varying signal orientations. While the impedance bandwidth (5.4% at 5.7 GHz and 5.2% at 9.5 GHz) is narrower compared to designs like that in [13] (31.4% and 8.8%), it is optimized for specific WLAN and X-band frequencies. The peak gains of 7.2 dBi and 5.8 dBi in this work are comparable to or exceed the gains in [13,15] for similar frequencies. Additionally, the presented design employs a Patch and Fabry–Pérot structure, which enhances performance while maintaining a broadside radiation pattern. In contrast, ref. [15] employs a dipole structure, leading to an end-fire pattern, less suitable for broad coverage applications. Overall, the combination of CP, consistent gains, and tailored design features positions this work as a significant improvement over existing designs, particularly for dual-band wireless communication applications.

## 5. Conclusions

This study highlights the successful development and application of a novel dual-band antenna array, demonstrating its potential for versatile and reliable wireless communication. The 4 × 1 antenna array achieves dual-band functionality through the strategic placement of parasitic elements near the radiating elements. This configuration enables efficient operation across two distinct frequency bands, supporting both WLAN and X-band applications.

A key innovation in the design is the incorporation of a Fabry–Pérot (FP) layer positioned above the antenna array at an optimized height. This layer enhances gain in the WLAN band by utilizing its resonant characteristics to improve antenna performance. In addition to boosting gain, the FP layer contributes to a more efficient radiation pattern, thereby improving the antenna’s effectiveness for WLAN communications.

Another critical design feature is the use of pentagon-shaped radiating elements, which play a vital role in achieving superior impedance matching. The geometric design of these elements ensures better alignment with the feed network, minimizing reflections and enhancing overall efficiency. This results in stable and reliable performance across the designated frequency bands.

Circular polarization is achieved across both the WLAN and X-bands using a sophisticated excitation mechanism that employs power dividers and λ/4 transformers. These transformers introduce 90-degree phase shifts to the excitation signals, a crucial step in generating circular polarization. CP is essential for maintaining robust and reliable wireless communication, as it ensures consistent performance regardless of signal orientation.

The integration of these advanced design features—the strategic use of parasitic elements, the Fabry–Pérot layer, pentagon-shaped radiating elements, and an advanced excitation method—validates the feasibility and effectiveness of the proposed antenna array. The study demonstrates the array’s ability to deliver enhanced performance, including improvements in gain, impedance matching, and polarization, across multiple frequency bands. These attributes make it a promising solution for modern wireless communication systems, particularly in applications requiring reliable and efficient multi-band operation.

## Figures and Tables

**Figure 1 micromachines-16-00203-f001:**
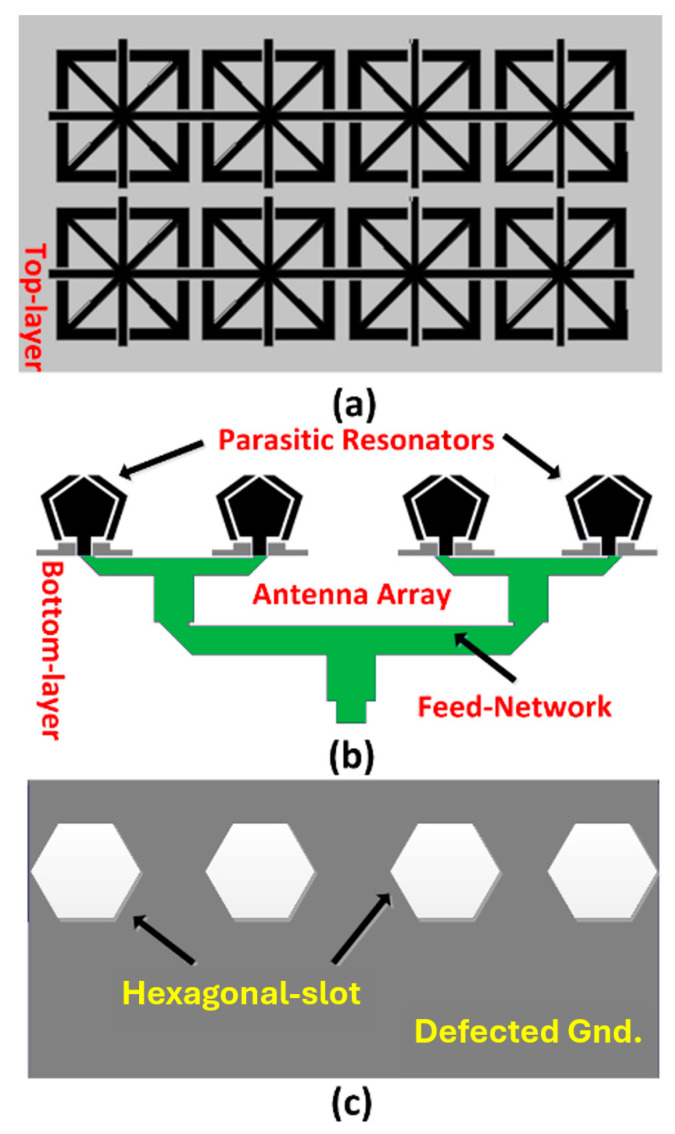
(**a**) Top layer with Fabry–Pérot array structure, (**b**) bottom layer of array with four parasitic resonators, and (**c**) the defected ground structure.

**Figure 2 micromachines-16-00203-f002:**
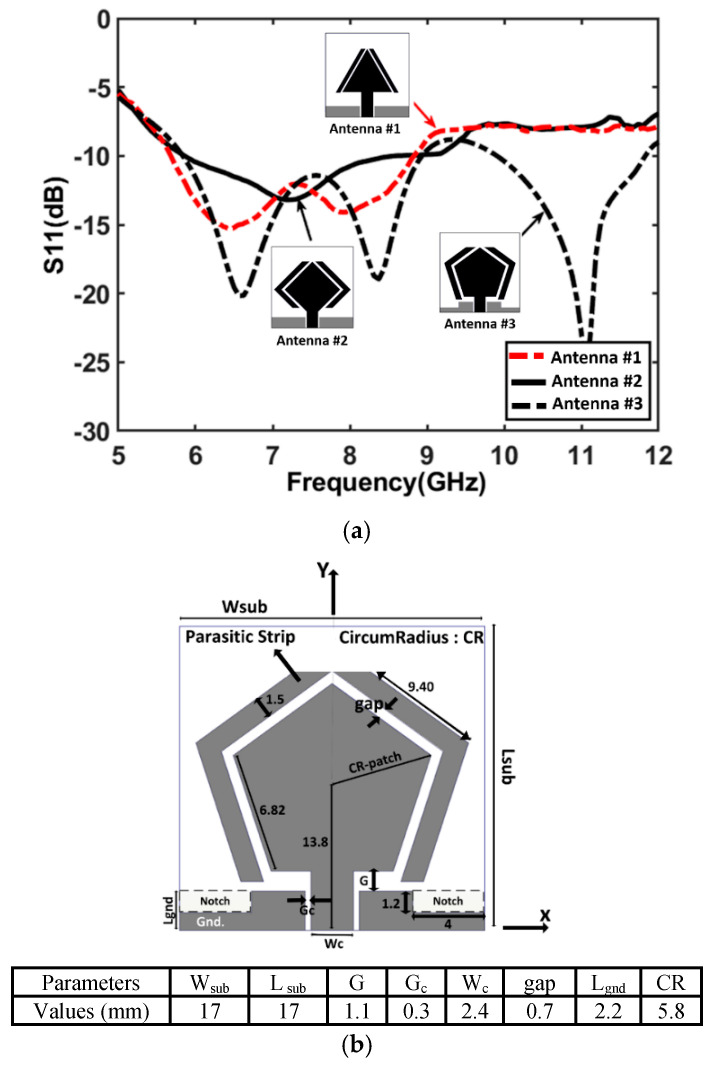
(**a**) Reflection coefficient response of the various radiating elements and (**b**) configuration of the proposed parasitic resonator used in the multi-band antenna array.

**Figure 3 micromachines-16-00203-f003:**
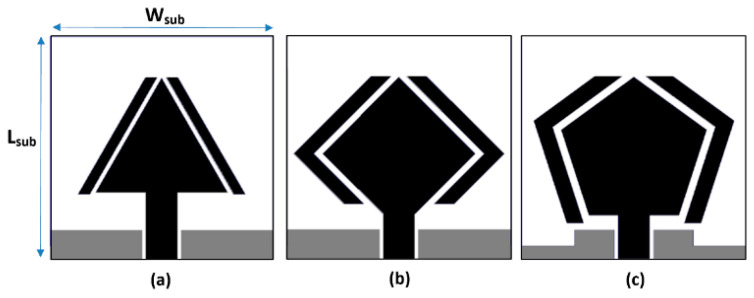
Steps used to realize the proposed antenna, (**a**) Triangular-shaped patch flanked by parasitic resonators with standard truncated ground plane, (**b**) Diamond-shaped patch flanked by conformal parasitic resonators with standard truncated ground plane, and (**c**) Pentagon-shaped patch flanked by conformal parasitic resonators and modified ground plane.

**Figure 4 micromachines-16-00203-f004:**
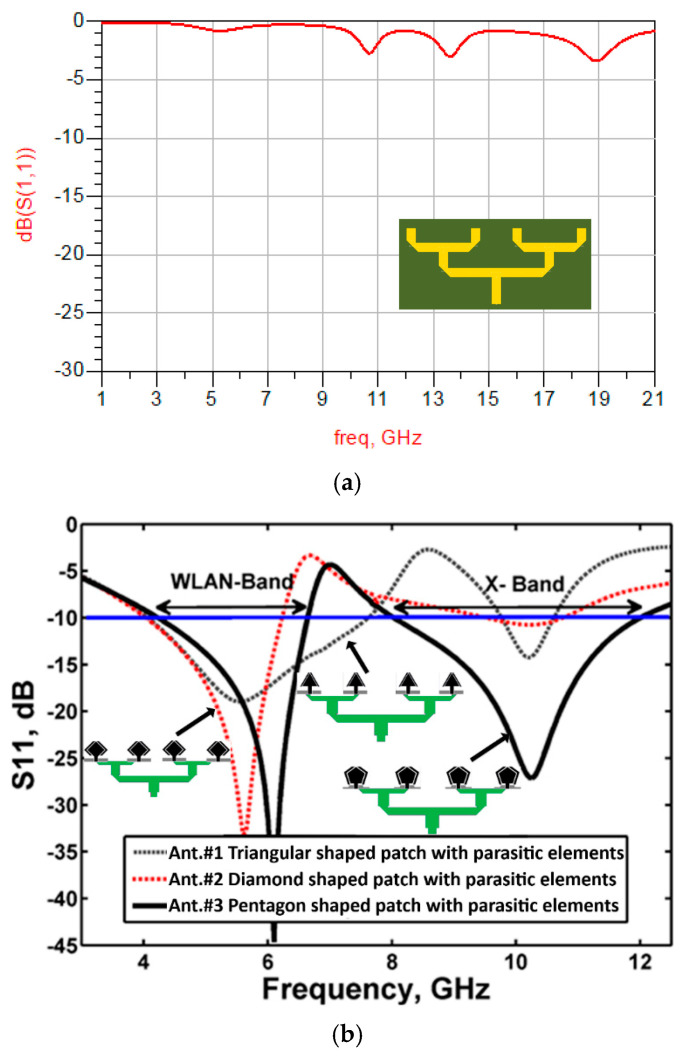
Simulated S_11_ response of the (**a**) Feeding network, and (**b**) 4 × 1 antenna arrays.

**Figure 5 micromachines-16-00203-f005:**
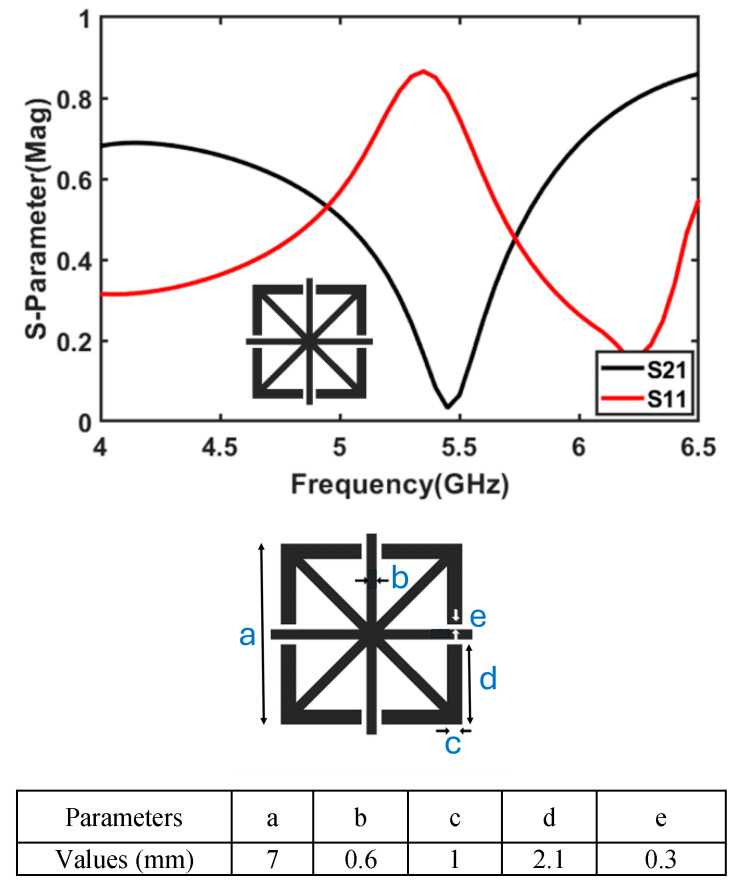
Scattering parameter response of the Fabry–Pérot unit cell.

**Figure 6 micromachines-16-00203-f006:**
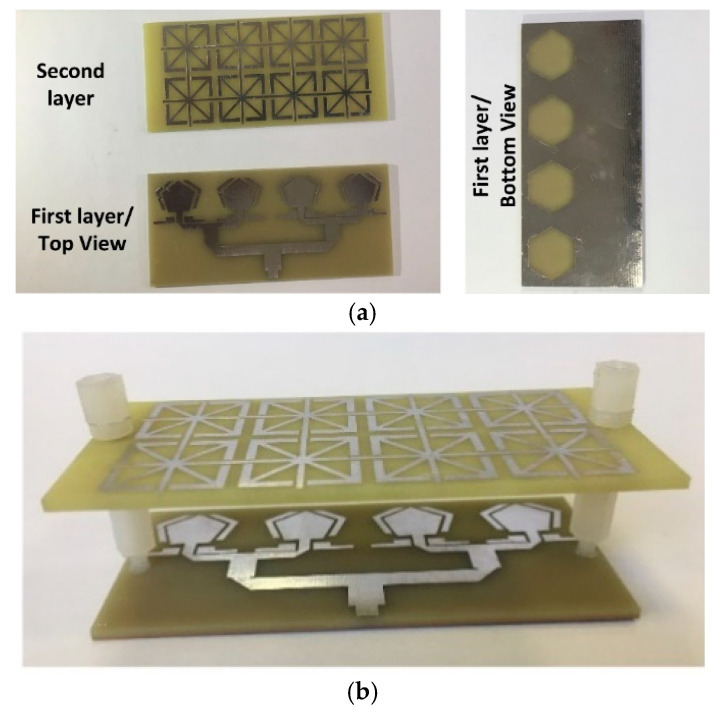
(**a**) Fabricated antenna array layers, (**b**) Assembled antenna array, (**c**) Measured and simulated reflection coefficient response of the proposed dual-band Fabry–Pérot antenna array, (**d**) Measured gain of the proposed FP antenna array with and without Fabry–Perot (FB), (**e**) Axial ratio of the fabricated FP antenna array, (**f**) Measured normalized radiation patterns for CP FP antenna array at 5.7 GHz, and (**g**) Measured normalized radiation patterns for CP FP antenna array at 9.5 GHz.

**Table 1 micromachines-16-00203-t001:** Performance comparison with previous antenna designs.

Ref.	Freq. (GHz)	Imp. BW (%)	Peak Gain (dBi)	Circular Polarization	Antenna Type	Rad. Pattern
[13]	3.5/28	31.4/8.8	2.1/8.8	No	Monopole/Slot	Broadside
[14]	3.7/28.5	3.4/4.5	8.9/19.2	No	Metasurface/FPRA	Broadside
[15]	3.5/28	20.7/20.5	7.1/11.3	No	Dipole/Dipole	End-fire
This work	5.7/9.5	5.4/5.2	7.2/5.8	Yes	Patch/Fabry-Pérot	Broadside

## Data Availability

The data presented in this study are available on request from the corresponding author.

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
