# Peer review of "Dual-Band Multi-Layer Antenna Array with Circular Polarization and Gain Enhancement for WLAN and X-Band Applications"

_micromachines, 2025, doi:10.3390/mi16020203_

Round 1
Reviewer 1 Report
Comments and Suggestions for Authors
The authors, Bal S. Virdee et al., have conducted a comprehensive study on a multi-layer, dual-band antenna array designed for WLAN and X-band applications. Before the manuscript can be considered for publication, I recommend the following revisions:
1. There is a typo error in the legend of Figure 1, and it should be (c) instead of (b).
2. In Figure 3(a), could the authors provide a more detailed explanation of why S11 ≤ -10 dB was chosen to calculate the impedance bandwidth?
3. Figure 3(b) illustrates numerous geometrical parameters for Antenna #3 that influence its performance. How were these parameters optimized? I suggest the authors discuss each parameter individually and analyze its impact on antenna performance.
4. Could the authors elaborate further on the rationale behind designing the Fabry-Pérot unit cell to resonate at 5.4 GHz?
5. In Figure 6(d), the measured gain of the proposed Fabry-Pérot (FP) antenna array with and without the Fabry-Pérot structure shows minimal differences in the X-band. Could the authors explore ways to further enhance the gain by optimizing the design of the Fabry-Pérot unit, such as adjusting its resonance frequency?
6. I am curious about the impact of the separation distance between the first and second layers on antenna performance. Could the authors provide additional discussion on how this distance affects the S11 performance?
Author Response
Thank you for your feedback.
1. There is a typo error in the legend of Figure 1, and it should be (c) instead of (b).
Author reply: Thank you for spotting this error. It has bee corrected in the revised version.
2. In Figure 3(a), could the authors provide a more detailed explanation of why S11 ≤ -10 dB was chosen to calculate the impedance bandwidth?
Author reply: S11 ≤ -10 dB threshold defines a frequency range where the antenna is considered "well-matched" for efficient power transfer without significant mismatch losses. We have clarified this in the revised paper.
3. Figure 3(b) illustrates numerous geometrical parameters for Antenna #3 that influence its performance. How were these parameters optimized? I suggest the authors discuss each parameter individually and analyze its impact on antenna performance.
Author reply: We have clarified this in the revised paper.
4. Could the authors elaborate further on the rationale behind designing the Fabry-Pérot unit cell to resonate at 5.4 GHz?
Author reply: The Fabry-Pérot (FP) unit cell is specifically designed to resonate at 5.5 GHz to improve the antenna array's performance in the WLAN frequency band, with a focus on enhancing gain and maintaining a consistent radiation pattern. We have clarified this in the revised paper.
5. In Figure 6(d), the measured gain of the proposed Fabry-Pérot (FP) antenna array with and without the Fabry-Pérot structure shows minimal differences in the X-band. Could the authors explore ways to further enhance the gain by optimizing the design of the Fabry-Pérot unit, such as adjusting its resonance frequency?
Author reply: Our paper was to demonstrate proof-of-concept. However, to enhance gain in the X-band, the FP unit cells could be modified to support dual-resonant behaviour, with a secondary resonance tuned to a frequency within the X-band. This would enable the FP structure to improve gain at both WLAN and X-band frequencies. Additionally, altering the geometry of the FP unit cells could increase their effectiveness at shorter wavelengths. We have clarified this in the revised paper.
6. I am curious about the impact of the separation distance between the first and second layers on antenna performance. Could the authors provide additional discussion on how this distance affects the S11 performance?
Author reply: We have clarified this in the revised paper.
Reviewer 2 Report
Comments and Suggestions for Authors
This manuscript introduces a dual-band circular polarized antenna array. The detailed comments are listed as follows:
1. The analysis of the antenna element design is too simple. Please give more principles and analytical results to demonstrate why the proposed antenna element has a wider bandwidth than the reference antennas.
2. Please give the simulated results of the feeding network without antennas. These results are critical to the performance of the entire antenna array.
3. Why the hexagonal defective ground structure is used? What effect this has on the performance of the antenna array? The manuscript does not give a reason.
4. Why the Fabry-Pérot layer of the 2 × 4 unit is used instead of the larger scale, and the position is not located directly above the center of the antenna array? This will have a great impact on the transmission phase.
Author Response
Thank you for your feedback.
1. The analysis of the antenna element design is too simple. Please give more principles and analytical results to demonstrate why the proposed antenna element has a wider bandwidth than the reference antennas.
Author reply: We have clarified this in the revised paper. The purpose of this paper is proof-of-concept.
2. Please give the simulated results of the feeding network without antennas. These results are critical to the performance of the entire antenna array.
Author reply: We have now included the response of the feed network in Fig. 4(a).
3. Why the hexagonal defective ground structure is used? What effect this has on the performance of the antenna array? The manuscript does not give a reason.
Author reply: Surface waves can cause interference, leading to higher reflections and reduced radiation efficiency. To mitigate these effects, we have introduced hexagonal slots in the ground plane to disrupt the surface waves through periodic discontinuities that limit their propagation. Additionally, in closely spaced antenna arrays, mutual coupling between elements can degrade overall performance. The hexagonal defective ground structure (DGS) reduces this coupling by altering the ground current distribution. As a result, the slots enhance the stability of the radiation pattern, reduce unwanted reflections, and lower cross-polarization, ultimately improving directional radiation performance. We have clarified this in the revised paper.
4. Why the Fabry-Pérot layer of the 2 × 4 unit is used instead of the larger scale, and the position is not located directly above the center of the antenna array? This will have a great impact on the transmission phase.
Author reply: A 2 × 4 unit-cell array is used instead of a larger FP structure to maintain compactness and avoid excessive weight or complexity in the antenna system. Enlarging the Fabry-Pérot layer may improve gain marginally but can introduce mechanical and spatial challenges, making it impractical for compact communication devices.We have clarified this in the revised paper. The purpose of this paper is proof-of-concept. The Fabry-Pérot layer is positioned slightly off-centre rather than directly above the centre of the antenna array to account for optimized field distribution and constructive interference at the operating frequencies. Positioning the Fabry-Pérot structure in this manner prevents destructive interference, optimizing gain and impedance matching without significantly affecting performance at the target frequencies. The transmission phase is influenced by the Fabry-Pérot structure's position relative to the antenna array. Misaligned or poorly positioned Fabry-Pérot layers can lead to phase distortion and destructive interference, which can degrade radiation performance. However, when the position and gap (typically 0.5λo) are optimized, the Fabry-Pérot structure enhances phase coherence, allowing for constructive interference at the target frequencies. We have clarified this in the revised paper.
Round 2
Reviewer 1 Report
Comments and Suggestions for Authors
The authors have addressed my concerns. I recommend the revised manuscript for publication.
Reviewer 2 Report
Comments and Suggestions for Authors
I have no further comments.